# Cooperative root graft networks benefit mangrove trees under stress

Alejandra G. Vovides [1✉], Marie-Christin Wimmler [2], Falk Schrewe[2], Thorsten Balke [1], Martin Zwanzig [2], Cyril Piou [3], Etienne Delay [4], Jorge López-Portillo [5] & Uta Berger[2]

The occurrence of natural root grafts, the union of roots of the same or different trees, is common and shared across tree species. However, their significance for forest ecology remains little understood. While early research suggested negative effects of root grafting with the risk of pathogen transmission, recent evidence supports the hypothesis that it is an adaptive strategy that reduces stress by facilitating resource exchange. Here, by analysing mangrove root graft networks in a non-destructive way at stand level, we show further evidence of cooperation-associated benefits of root grafting. Grafted trees were found to dominate the upper canopy of the forest, and as the probability of grafting and the frequency of grafted groups increased with a higher environmental stress, the mean number of trees within grafted groups decreased. While trees do not actively 'choose' neighbours to graft to, our findings point to the existence of underlying mechanisms that regulate 'optimal group size' selection related to resource use within cooperating networks. This work calls for further studies to better understand tree interactions (i.e. network hydraulic redistribution) and their consequences for individual tree and forest stand resilience.

[1] School of Geographical and Earth Sciences, University of Glasgow, Scotland, UK. [2] Institute of Forest Growth and Forest Computer Sciences, Technische Universität Dresden, Dresden, Germany. [3] CIRAD, UMR CBGP, INRAE, Institut Agro, IRD, Univ Montpellier, Montpellier, France. [4] CIRAD, UR GREEN, Montpellier, France. [5] Functional Ecology Network, Instituto de Ecología A.C., Veracruz, Mexico. ✉email: Alejandra.Vovides@glasgow.ac.uk

Natural root grafts, the physical connection of two roots belonging to different trees or a single individual tree, have been known about for more than 100 years[1] and are recognised in almost 200 tree species[2,3]. However, until the last decade, we had had little information about their ecological implications for tree interactions and forest stand dynamics. Between the 1950s and the 1990s, they were mainly regarded as a phenomenon of random occurrence or a threat to forest stands due to their role as vectors of pathogen transmission in forest stands[4] and the only long-term accepted consequence of grafting as a positive trait was increased mechanical stability[2,3,5]. Now, the common perception of forest dynamics being ruled only by competition and survival of the fittest is being challenged by the discovery of mycorrhizal networks and the re-evaluation of root grafts as vectors of positive interactions amongst trees[6–11].

Root grafts are functional when cambia and vascular tissues are fused[2]. Functional grafts can facilitate resource exchange[9] and promote growth[12] by mitigating the adverse effects of defoliation and budworm outbreaks[13,14] and by increasing the concentration of carbohydrates in shaded trees[9]. Still, even if grafts are non-functional, they can increase tree stability as compared to non-grafted trees by sharing anchoring systems[5]. The latter is particularly relevant for coastal wetland forests, which are regularly exposed to strong winds[15] and further characterised by shallow root systems due to anoxic sediments[3,16].

By enabling the exchange of water, carbon, and mineral nutrients, functional grafts can keep severed trees alive through the support of grafted neighbours[17]. Moreover, modelling approaches suggest that natural root grafts could explain short-range positive interactions that lead to large-scale fractal patterns in tree yield[18] that seem to predestine natural root grafting as a cooperative trait. However, our current knowledge is mainly based on the study of grafted pairs of trees, while spatially explicit field investigations are limited to small plots in terrestrial forests (mainly for logistic reasons)[1,10,19,20]. The functional ecology of root grafts in wetland forests, and the effect of environmental stress on network topology remains unexplored.

Until now, the study of root grafts has required the extensive excavation of root systems[4,12,19–21] and, often, decades to gather quality information[12]. The inclusion of environmental gradients to understand positive plant interactions and their ecological implications for community dynamics[22,23] is therefore very limited. However, mangrove forests, with their distinct environmental (i.e. elevational or inundation) gradients[23] and traceable shallow root system with pneumatophores (e.g., the pencil-like emerging roots of *Avicennia* spp.), offer an ideal model system to study the ecological role of root graft networks. *Avicennia germinans* L. dominates forests on hypersaline mudflats with limited tree diversity, and strong salinity gradients offer particularly satisfactory conditions to study physiological responses[24], tree architecture, and tree interactions[15,25,26].

These mangrove specificities provide ideal conditions to investigate in the field whether root grafting can benefit trees through the analysis of individual tree attributes and spatial root graft network structures along environmental gradients. They also represent challenges; for instance, annual growth patterns in tropical forests are unclear and, for mangroves, widely affected by local environmental conditions (i.e. rainfall, competition and salinity)[27], thus dendrochronological studies to assess tree and graft ages are unreliable, limiting our ability to disentangle the cause-effects of grafting on growth rates. In addition, the spatially explicit character of the presented work enabled an innovative mapping of tree networks in extensive forest areas, but this was at the expense of performing anatomical studies to check the functionality of the root grafts because the roots could and should not be excavated and destroyed for entire mangrove stands.

Instead, we comprehensively analyse stem-height ratios and resulting slenderness ratio of grafted and non-grafted trees in relation to competition pressure, we also study the changes in network structures along increasing environmental stress, which can inform on interaction patterns despite functional anatomy studies of grafts are lacking (i.e. emerging patterns of optimal group-size selection for cooperative groups[28,29]), and discuss the remaining uncertainties of the results due to the missing information on root grafts age and functionality.

By analysing allometric changes in whole stands and focusing in network structure along environmental stress gradients, we show that environmental stress controls the size of the groups formed by grafted trees. This suggests that root grafts could be an adaptive trait in trees that might also contribute to individual and forest resilience.

## Results and discussion

**Drivers of root grafting.** To understand the main drivers and consequences of natural root grafting in an *A. germinans* dominated forest, we focused on a seasonally hypersaline mangrove forest bordering the coast of the Gulf of Mexico (Fig. 1a). A steel rod root detection method developed to measure root length with minimal excavation[30] was modified to identify and map root graft networks in eight 900 m² forest stands (Fig.1c–d). We further related root graft frequency to biotic and abiotic variables, such as stem diameter, stand density and porewater salinity (see methods section). We also explored the height–diameter relationship of grafted and non-grafted trees with different neighbourhood asymmetries (i.e. competition pressure by neighbours; see Supplementary Methods, and Supplementary Figs. 1 and 2) and the relationships between the related network group attributes of group size and frequency to stand density and salinity.

The lowest porewater salinity was recorded in plot 1 (39.7 ± 1.5, mean ± SE) and the highest at plots 8 (58.62 ± 1.2 ppt) and 13 (58 ± 0.8 ppt) (Supplementary Table 1); *A. germinans* stand densities ranged from 300 to 900 trees ha$^{-1}$ (Supplementary Table 1). While overall root graft frequency ranged between 34 and 70% amongst plots (with a mean of 56.5 ± 4%), 76.9 ± 5% of the top-height trees were grafted (Supplementary Table 1). Top-height trees, defined as the 20% biggest trees in a stand (as per stem diameter, see methods section), are considered to have exploited resources to their maximum ability, and thus reflect the potential productive capacity of a stand[31]. The high frequency of grafting in the most dominant trees suggests that a shared root system provides essential advantages to the forest, either by optimizing resource exploitation or by increasing mechanical stability and windthrow resistance.

In a logistic regression (with an accuracy of prediction of 70.5%), to assess the drivers of root grafting showed, in line with previous studies[19,32,33], a higher probability of grafting with increasing tree stem diameter ($p < 0.0001$; Fig. 2a, Supplementary Table 2, Supplementary Data). With the addition of salt stress, however, the contribution of stem diameter to grafting probability decreased for stem diameters >20 cm (Fig. 2a; Supplementary Data; Supplementary Table 2). In upland forests, higher stand densities contribute to increased grafting probability due to reduced distances between neighbouring trees[32]. In our study, however, stand density in interaction with salinity significantly reduced grafting probabilities (Supplementary Table 2). This could be associated to higher resource limitations within saline environments. First, closer neighbours result in greater competition[34], while high salt stress additionally reduces resource availability and limits growth rates[24,34], leading to smaller stem diameters within the forest, and hence to lower probability to graft. Nevertheless, trees with smaller stem diameters had a

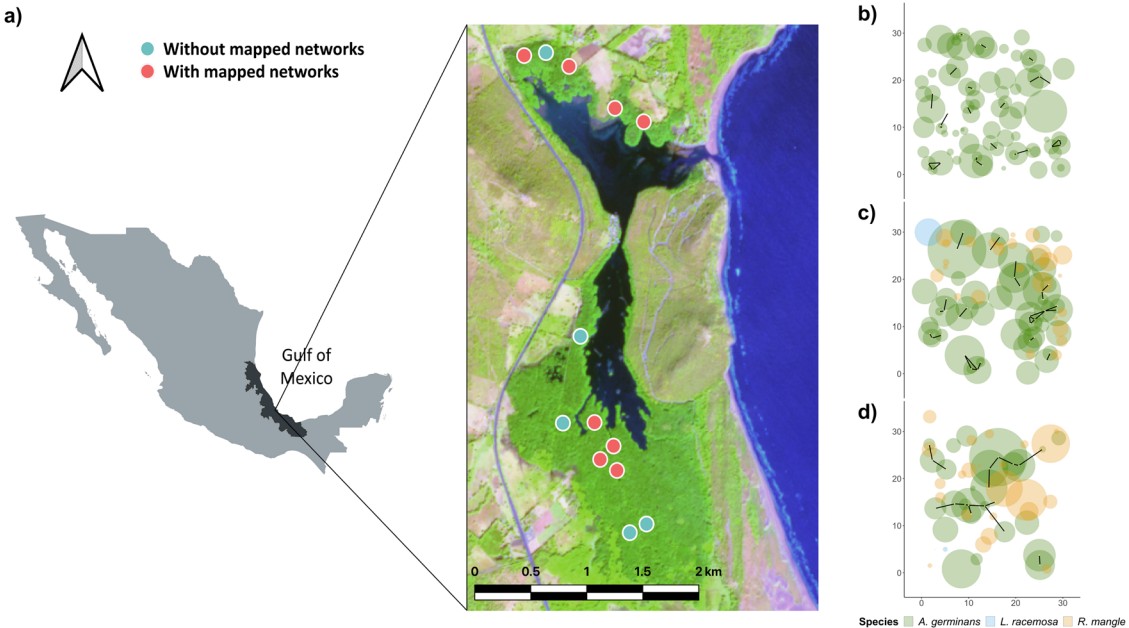

**Fig. 1 Study site and root network maps located on the central coast of the Gulf of Mexico. a** La Mancha lagoon, surrounded by mangrove vegetation (@Copernicus Sentinel Data [2020]), contains 13 permanent plots (red and blue points) used for vegetation monitoring. Root grafts were mapped within eight of these stands (red points). **b**–**d** show root-grafted tree network maps representative of sites with: **b** high, **c** medium and **d** low salinity and stand densities. The yellow dots depict stem positions; the coloured circles (green, peach and blue) are the tree crowns of black mangrove (*Avicennia germinans*), red mangrove (*Rhizophora mangle*) and white mangrove (*Laguncularia racemosa*), respectively; and the black lines represent graft connections.

higher probability of grafting at higher salt-stress plots and marginally lower grafting probabilities at increasing stem diameters (Supplementary Table 2). The highest proportion of grafting was recorded for plots with the highest stand densities and salinities (Supplementary Fig. 3), suggesting that salt stress has direct control over root grafting, although overall tree size decreases with increasing salt stress.

**Allometric patterns of grafted and non-grafted trees**. A generalised additive mixed model used to explore the effect of grafting on tree size, demonstrated that grafted trees are generally taller than non-grafted trees ($p < 0.01$, Fig. 2b; Supplementary Data). Although neighbourhood asymmetry did not influence tree height ($p = 0.37$ and $p = 0.28$, for grafted and non-grafted trees, respectively) (Supplementary Fig. 4a–b), grafted trees have a more linear relationship between stem diameter and height ($p < 0.001$; Fig. 2b; Supplementary Data; Supplementary Fig. 4c) contrasted with the markedly reduced rate in height increase for non-grafted trees with stem diameters >10 cm ($p < 0.001$; Fig. 2b; Supplementary Data; Supplementary Fig. 4d). The model had a strong coefficient of regression (adjusted $r^2 = 0.72$), explaining 81% of the deviance and had no overdispersion (Supplementary Fig. 5), suggesting grafts provide a benefit to trees. This could be related either to a potential increased growth rate[9,19] if grafts are fully functional, or to increased mechanical stability related to an extended area for anchorage[5].

The height to stem ratio (so called slenderness) is an allometric trait[35] that determines mechanical stability. Very slender trees are more vulnerable to windthrow, while low slenderness coefficients increase wind resistance[37]. Slenderness varies throughout tree development; younger trees invest in height growth before girth as a result of competition for light, which increases their risk of mechanical failure[37]. As they reach the canopy, more resources are invested in stem girth, conferring higher mechanical stability and resistance to windthrow[35–37]. Further a multiple linear regression to assess changes in slenderness along the range of

stem diameters (Fig. 2c; Supplementary Data), revealed a significant increase in slenderness for non-grafted trees that are subject to higher competition pressure ($p = 0.02$, adjusted $r^2 = 0.54$, Supplementary Table 3). However, there was also a significant, but weak negative interaction between stem diameter, grafting condition and neighbourhood asymmetry ($p = 0.01$). As non-grafted trees increase in diameter, their slenderness decreases more rapidly than it does for grafted trees (Fig. 2c; Supplementary Data). Hence, at larger stem diameters, a grafted tree is more likley to have higher slenderness than a non-grafted tree (Fig. 2c). This points towards potentially increased mechanical stability for dominant grafted trees subject to greater wind exposure[37]. This finding opposes a report of lower slenderness for grafted hybrid poplar clones[10], which could be explained by the comparisson being made between the overall means of grafted and non-grafted trees, but excluding the effect of stem diameter[10].

**Network formation**. Trees can benefit from functional root grafts through the increase of foraging area via communal root systems[2,38]. Root networks could also mitigate salinity-induced physiological drought through water redistribution between stems[14,17]. Further, shared carbohydrate pools could improve tree responses to both abiotic and biotic stress[7,8]. These factors likely contribute to the dominance of the forest canopy by grafted individuals. We challenged this hypothesis through the analyses of network topologies along the salt-stress gradient, where the lack of resource exchange would result in random network formation patterns, and similar network topologies along the stress gradient.

In forests, the location of individual trees is fixed after their establishment, and network formation is determined by physical, genetic and size proximity[32], limiting any preferential attachment processes. Although the root networks in our study might fit a scale-free power-law distribution ($p = 0.21$; Fig. 3a), that is, they might possess patterns of continuous growth and preferential attachment[39], we found no significant power of determination to

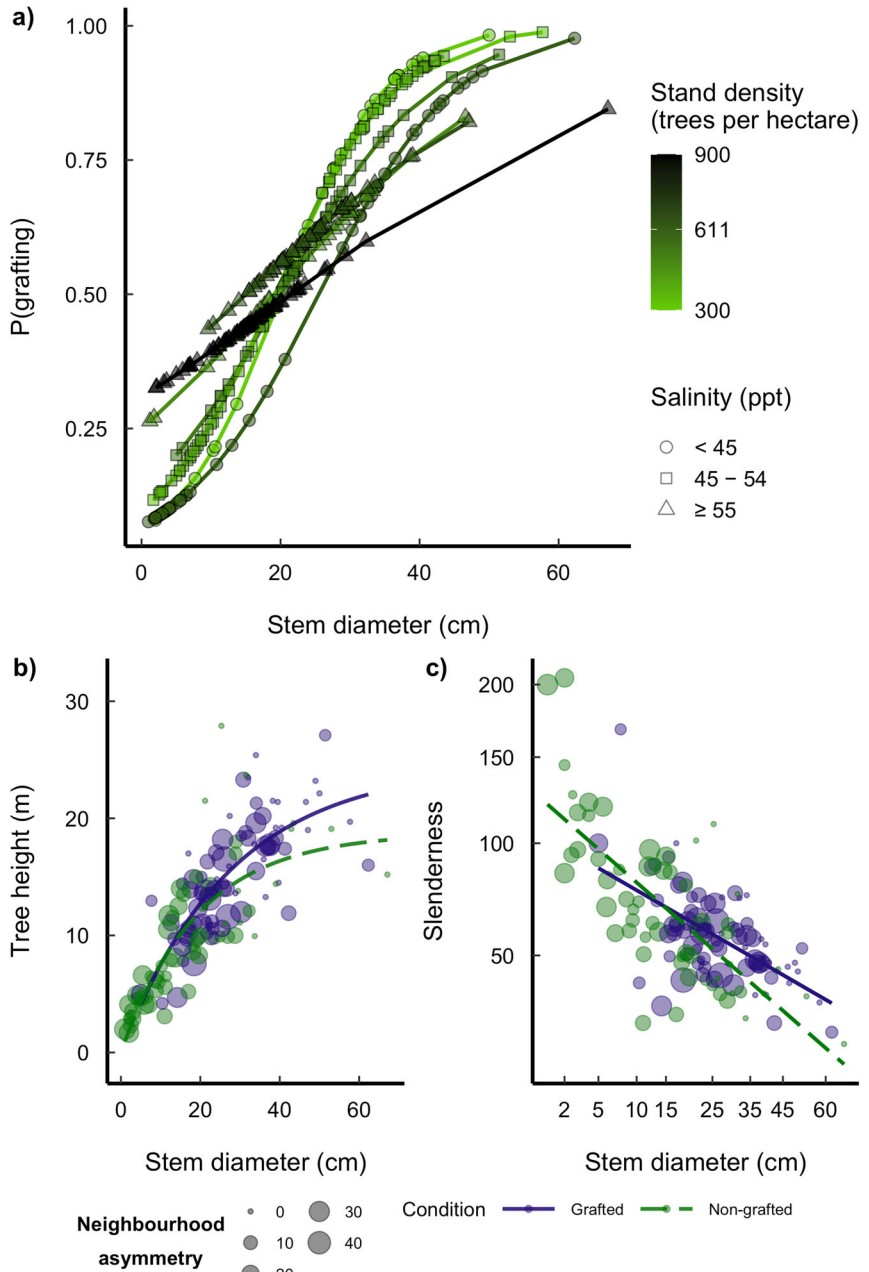

**Fig. 2 Probability of rafting and allometric differences between grafted and non-grafted trees. a** Logistic regression testing the main drivers of root grafting. Although, for all cases, the probability of grafting increases with increasing stem diameter, at higher salt stress (symbols) and total density values (colour scale), trees graft at smaller stem diameters, and the probability curve rises earlier and is steeper ($N = 324$); **b, c** Allometric patterns in relation to neighbourhood asymmetry (circle size) and grafting ($N = 141$); **b** Non-linear relationship between stem diameter and tree height showcasing stem diameter–height allometric curves that are steeper for grafted (purple) trees compared with non-grafted (green) trees. There is a predominance of grafted trees with stem diameters >20 cm, whereas the tallest trees are either grafted or have smaller values of neighbourhood asymmetry; **c** Sharp decrease in slenderness coefficient with increasing stem diameter for non-grafted trees, at higher stem diameters grafted trees are more slender.

reject random network formation when comparing the power-law distribution to log-normal ($p = 0.99$), Poisson ($p = 0.93$), or exponential distributions ($p = 0.99$; Fig. 3a; Supplementary Data). In this context, grafting could be a random process;[2,19] however, the distribution of the node degree (number of trees connected to a given tree via root grafts), the group frequency distribution and group size along the stand-density gradient (Fig. 3b–c; Supplementary Data) point to underlying mechanisms that select for optimal group size in cooperative groups[39].

Most of the grafted trees were connected to one (61%) or two (29%) individuals, while connections to four partners were rare

(1%). Hence, the node degree's relative frequency is smaller with increasing node degree (Fig. 3a). Additionally, the average node degree is negatively correlated to the frequency of grafted trees ($r^2 = 0.93$; $p < 0.001$; Fig. 3b; Supplementary Data), and stand density (adjusted $r^2 = 0.40$; $p = 0.06$; Fig. 3b; Supplementary Data), where the marginal correlation was attributed to the high density and low grafting frequency of trees in plot 3 (see Supplementary Table 1), for which 51% of *A. germinans* trees had stem diameters <15 cm, as compared to most similar sites in salinity and located near plot 3 (plots 1 and 2, with 22 and 28% of trees with stem diameters <15 cm, respectively). When considering plot 3 as an

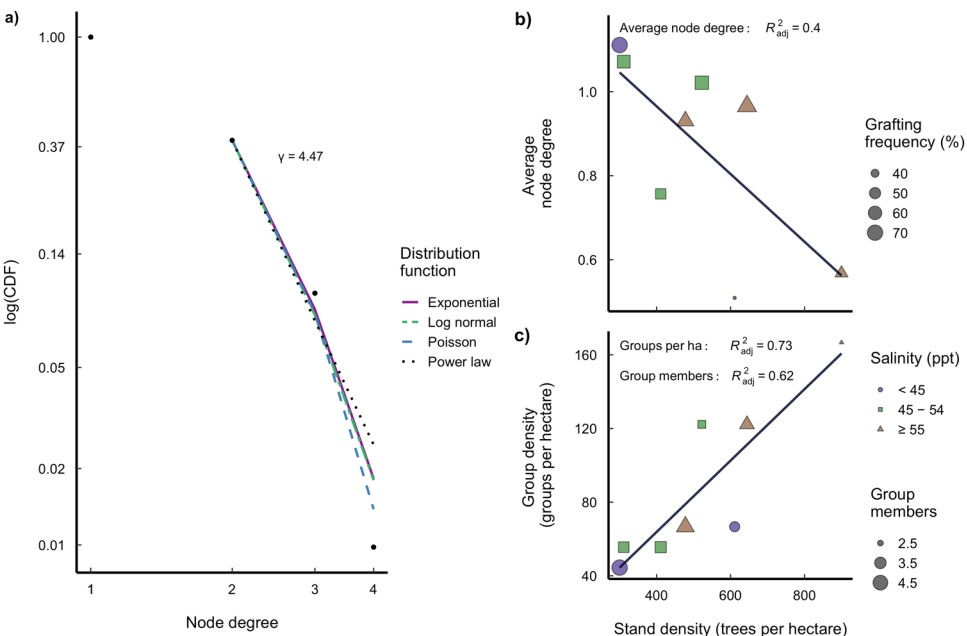

**Fig. 3 Root graft network attributes. a** Comparison of log-transformed cumulative distribution functions (CDF). The number of direct connections for a given tree (node degree) shows that root graft networks might fit a scale-free power-law distribution with a slope $\gamma = 4.47$. But with a min. node degree of 2, the Poisson, log-normal and exponential distributions cannot be discarded. **b** Linear regression showing a reduction on the average node degree of the networks along the *A. germinans* stand-density gradient and grafting frequency (%). Shape and colour depict the salinity range of the plots (purple circles <45 ppt, green squares 45–55 ppt and brown triangles ≥55 ppt) and the size of each shape indicates the stand's grafting frequency. **c** Linear regression showing an increasing number of groups of grafted trees per hectare with significant reduction in the mean group members (shape size), this is, the number of trees forming a group.

outlier, removing it from the analysis returned a significant negative correlation between average node degree and stand density (adjusted $r^2 = 0.49$, $p = 0.04$). Consequently, the average node degree was highest in the plot with the lowest stand density and highest grafting frequency (Fig. 3b; Supplementary Data). Likewise, as *A. germinans* stand density increased, the average number of trees forming groups became smaller (adjusted $r^2 = 0.62$; $p = 0.01$; Fig. 3c; Supplementary Data), whereas the frequency of groups increased (adjusted $r^2 = 0.73$; $p < 0.01$; Fig. 3c; Supplementary Data). This is in line with network theory findings of cooperative interactions increasing with environmental stress[28,40]. Such interactions, however, do not come without costs. It costs each cooperating individual to provide a benefit to its neighbours, and to be selected as an adaptive trait within a population, the net gain of the cooperative trait should be greater than its cost[40].

There is evidence that in unweighted networks, selection favours cooperation when the benefit–cost ratio $\left(\frac{b}{c}\right)$ exceeds the average number of neighbours ($k$) (i.e. node degree): $\left(\frac{b}{c}\right) > k$. Thus, most cooperative groups tend to have few members[28,29,40] and higher probabilities of direct reciprocity (i.e. pairwise ties)[29]. The average node degree in the mangrove root networks we studied was smaller at sites with high salinity and high stand density compared with low salinity and medium stand density (Fig. 3b; Supplementary Data). However, stand densities were similar in low-and high-stress environments. As the cost of cooperation increases under stressful conditions[28], assuming that a tree receives constant benefit from its cooperating neighbours, the critical $\left(\frac{b}{c}\right)$ ratio decreases with increasing stress. Thus, larger tree groups are not selected under situations of limited resource availability. Most of the grafted mangrove groups (73%) consisted of only two or three members. However, of the groups that included more than two trees, 72% had no more than the minimum required number of connections for a cooperative

system (each individual had at least one connection for cooperation within its group members). This supports the hypotheses that functional root grafts can only be maintained if there is a long-term payoff for all group members and underlying mechanisms selected for optimal group size in root networks. These network topologies provide preliminary evidence that root networks are the result of cooperation, enabling small groups to perform better under stress[28,40], and suggests that spatial resource limitation affects the network structure by modulating the number of interacting individuals and, potentially, the magnitude of their interactions[41].

**Concluding remarks.** Our findings represent indirect evidence of positive interactions between trees via root grafts, which are reflected on tree allometry and network attributes. Nevertheless, it is noteworthy to highlight that quantifying size, age and functionality of grafts is still necessary to assess the relative contribution of functional grafts to resource transfer[9–11], as they are detrimental to disentangle the cause-effects of grafting on tree growth and fitness. Age of grafts, as well as their size affect their functionality and thus, their ability to transfer resources between trees[9]. Given that such factors fell out of the scope of this study, inferences on cause-effects of grafts on tree size are not possible here, we can only discuss that the increased slenderness of grafted trees at stem diameters >25 cm could be related to increased mechanical stability[10,42], potentially contributing to their dominance at the forest canopy level, but might not necessarily be related to resource transfer or increased growth rates between grafted trees. While the observed network structure changes with increasing environmental stress suggest functional grafts are present in the system, their quantification and contribution to resource transfer withing groups is needed for a mechanistic understanding of the processes regulating group sizes and the

ecological implications of networks for trees and forest stand performance.

Further studies on the ecological implications of network formation via root grafts (i.e. the implications of root grafting for water-use efficiency, hydraulic redistribution and nutrient exchange), will contribute to a greater understanding of the trade-off between positive and negative interactions. That is, positive interactions confer ecological advantages that help overcome harsh environmental conditions, although the cost could include a higher risk of pathogen transmission. More broadly, our findings widen the path opened by Kropotkin's (1902) *Mutual Aid: A Factor of Evolution*[43], which was largely forgotten during the 20th century.

## Methods

**Study site**. The study site is located on the central coast of the Gulf of Mexico (GoM) at the La Mancha lagoon (Fig. 1a) at 19°33′–19°36′ N; 96°22′–96°24′ W[44]. The surface of the lagoon's waterbody covers 135 ha and is surrounded by 300 ha of mangrove forest[45]. Annual precipitation in the area ranges between 1200 and 1500 mm, and the annual mean temperature is 25 °C[46]. Freshwater and marine water inputs into the lagoon come from extreme opposite directions: the connection to the GoM is located in the northern extreme, while a main riverine input is located in the southern extreme. This creates a year-round salinity gradient that increases northward, regardless of seasonality[15,47,48]. This is also reflected in the zonation of the mangrove species, where the northern and most saline environments are mainly represented by *A. germinans* with minor *Rhizophora mangle*, which gradually pass into mixed stands co-dominated by *R. mangle* and *A. germinans* or *A. germinans* and *Laguncularia racemosa* towards the southern end of the lagoon[47,49].

Within this mangrove forest, 30 × 30 m permanent plots established in 2010 are arranged along the salinity gradient. They are all oriented to true north and located equidistant from the main waterbody. The seven plots selected for this study are located at increasing distances from the lagoon's inlet to the GoM (between 500 and 3000 m) to capture the salinity gradient along the lagoon. A new plot within the highest salinity range was established in 2017 to include a site with a stand density similar to other sites but with contrasting salinity. For the pre-established plots, existing tree parameters were recovered from a publicly available database[15,50], including a unique ID, species, x- and y-axis positions in the plot, stem diameter at 130 cm from the soil surface ($D_{130}$) and height ($H$). For the newly established plot, the same tree parameters were measured using a laser rangefinder (Laser Rangefinder Forestry Pro 550; Nikon Vision Co., Ltd, Tokyo, Japan), and tree positions were determined using a compass and the rangefinder following standard forestry procedures[51]. A total of 482 trees were recorded for all plots.

For each plot, during April and September 2017, two pseudo-replicate porewater samples were collected from each corner and the middle of the plots from 20 cm below the ground surface using a custom-made porewater extractor[52] and immediately analysed for pH, salinity, temperature and redox potential (Ultrameter II; Myron L Company)[53].

**Root graft data collection**. A non-destructive method was used to detect the potential location of root grafts using a portable Doppler ultrasound probe (DU; SonoTrax Basic; Edan Instruments GmbH, Hessen, Germany) and a set of steel rods. The mangrove roots were gently located with steel rods with the DU probe placed on the tree stem. Following an adapted method originally developed to measure the woody root extensions of *A. germinans*[30], the probe was then gradually moved from the stem to the consecutive rods in contact with the target root. Each tree was examined following the consecutive order of the tree tag numbers within the plots by assessing their grafting to all immediate neighbours.

Placing the DU on a tree stem collar ring, a steel rod was used to probe the soil to shallow depths, and an amplitude monitor indicated when a root belonging to the stem was touched. Leaving this first steel rod in contact with the root, a second rod was used to further probe close to the first rod in the assumed direction of the course of the root until another positive signal was obtained. The interchangeable waterproof probe of the DU was then attached to the second steel rod, having been proofed to be in contact with the initial root, and the process repeated until either the root was too deep or too thin to be followed or led to another tree. In the latter case, the probe was held on the second tree stem and the last verified steel rod was used to again probe until another positive signal was returned by the DU from the second stem. The DU-located root graft was then verified by localised excavation of each target tree's neighbour. Although we were unable to verify false negatives, we calculated a 6% probability of finding false-positive connections (i.e. 12 false positives out of 200 connections detected), all identified false positives were treated as non-grafted trees. We did not have any means to evaluate false-negative rates.

Using this method, all *A. germinans* tree's (376) root systems were followed during April and May 2017. These were mapped and used to determine the grafted network topology: node degree (number of direct connections for each tree),

number of groups of grafted trees and mean group size (number of individuals within a group).

To estimate the pressure each tree receives from its neighbours, an index of neighbourhood asymmetry was calculated as a function of the size and distance of all neighbouring trees ($trees_j$) within a 5 m radius of the target tree ($tree_i$; see Supplementary Methods for computation details) regardless of their species. A large index of neighbourhood asymmetry implies that the neighbours are large and in close proximity, potentially exerting higher competition pressure on a target tree than a small neighbourhood asymmetry would. The 5 m radius was chosen because it had been previously identified as the optimal radius for detecting the responses of trees to its neighbours at the same study site[15]. Neighbourhood asymmetry was only calculated for trees where their complete neighbourhood was within the limits of the sampling plots (183 trees) to avoid biased neighbourhood asymmetry sizes related to incomplete information for neighbouring trees located outside a plot.

**Statistics and reproducibility**. Both the density of the target species *A. germinans* and the total stand density (including *A. germinans*, *R. mangle* and *L. racemosa*) were calculated as the number of trees per hectare. The replicate porewater salinity values for each sampling point were averaged, and the resulting five salinity values were used to estimate a mean plot salinity, including the standard error. The proportion of grafted trees at each plot was calculated as the number of *A. germinans* grafted trees divided by the total number of grafted *A. germinans* trees in the stand. The top-height trees at each stand (the 20% biggest) were selected as per stem diameter because it was measured in the field and is considered more accurate than tree height, which was estimated through stem diameter measurements[51].

Logistic regression was implemented using a generalised mixed effects model to assess the probability of grafting as a function of stem diameter, total stand density and salinity. The model included site identity as a random effect and stem diameter, site salinity and total stand density as fixed effects after assessing the autocorrelation between response variables (Supplementary Fig. 6) and all intra- and cross-level interactions between stand density and salinity. All the variables were z-transformed using the mean and standard deviation of each variable across all sites. To additionally estimate confidence intervals of the odd ratios based on stem diameter, for smaller trees (assumed to be 1 SD below the mean) we added 1 SD from the z-transformed value of stem diameter[54], and accordingly, we subtracted 1 SD from the z-transformed value of stem diameter for higher stem diameter trees[54].

To explore the effect of root grafting and neighbourhood pressure on tree allometry, in the generalised additive mixed effects model (GAMM), salinity and condition were included as fixed effects (cyclic cubic regression spline), neighbourhood asymmetry and stem diameters were included as smooth terms with smooth functions (Duchon spline) and the sampling plot was included as a random effect. The best model explaining tree height was selected using a minimal Akaike information criterion value following a stepwise removal of non-significant response variables ($N = 141$ single-stem *A. germinans* trees with a computed neighbourhood asymmetry).

In the existing database of tree parameters[50], multiple-stem trees are recorded following the traditional convention of summing the diameters of each stem but by measuring only the height of the tallest stem[26], leading to inaccurate diameter-height allometry. To avoid biased results when relating stem diameter to stem height and the probability of root grafting, multiple-stemmed trees (52 trees) were neither included in the logistic regression, nor GAMM, (for which we also excluded trees that did not have their full neighbourhoods inside de plots), resulting in a final number of trees of 324 and 141 included in the logistic regression and GAMM, respectively.

To further assess the effects of root grafting on tree slenderness ratio (an allometric trait that modulates mechanical stability), a linear model was used to evaluate the variations in the slenderness coefficient on the 141 single-stem *A. germinans* trees for which neighbourhood asymmetry was computed. To normalize the data, we performed a square root transformation of both the slenderness index and stem diameter. The model included slenderness as response, and an interaction term between grafting condition stem diameter and neighbourhood asymmetry. The final model was plotted back transforming the x- and y-axis to the original values of stem diameter and slenderness for simplicity of figure presentation (Fig. 2c).

Network parameters (node degree, number of groups per hectare and group size) were used to assess random network formation by comparing the probability of networks having a scale-free power-law distribution with random process distributions (i.e. log-normal, exponential, and Poisson). Scale-free networks do not occur randomly because a relative change in one node results in a proportional change in another node. Scale-free power-law distributions indicate the continuous expansion of networks and preferential attachment, where new nodes are constantly added and previously well-connected nodes are more likely to acquire new connections[39,55]. We then related network node degree, group size and frequency (number of trees grafted within groups and frequency of groups per hectare, respectively) to stand density and site salinity using simple linear regressions. All 376 *A. germinans* trees, including multi-stemmed trees, were included in this analysis, as these tests did not use any tree allometric attributes, such as tree height or stem diameter. For the linear regression assessing the relationship between average node degree and forest stand density; however, we

removed plot 3 from the analysis, as it has an atypical high stand density and low graft frequency that can be explained by the overall high density of small trees (51% of all *A. germinans* trees had stem diameters <15 cm).

All the statistical analyses were conducted using R programming language[56]. Specifically, we used the lme4[57], DHARMa[58], and gamm4[59] packages for the logistic regression and the GAMM construction and diagnosis. For the network analyses, we used igraph[60] to estimate the node degrees and PoweRlaw[61] to explore the distribution. All figures presented were developed using ggplot2[62].

**Reporting summary**. Further information on research design is available in the Nature Research Reporting Summary linked to this article.

## Data availability

Data on allometric attributes of the trees (i.e. height, stem diameter and position in stand) are publicly available at https://doi.org/10.5525/gla.researchdata/657/ [50]. All data that support the findings of this study are publicly available at our GitHub repository (https://github.com/mcwimm/GRINanalysis/tree/master/data).

## Code availability

All the coding resources generated for this study are publicly available at our GitHub repository (https://github.com/mcwimm/GRINanalysis).

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

## Acknowledgements

This paper is dedicated to the memory of Adán Vez Lira, who died on the 8th of April 2020 defending the natural resources of the La Mancha community. This research was financed by the Volkswagen Foundation (Volkswagen Stiftung, Project No. 94 844) and the MOMENTS project (NERC, UK grant No. NE/P014127/1). Support by the Association of Friends and Sponsors of TU-Dresden e.V. and INECOL was provided to F.S. (GFF, grant No. 61/2017 and INECOL, project No. 00016, respectively). Editing support and open access funds were provided by the Research and Innovation services of the University of Glasgow.

## Author contributions

A.V., J.L.P., F.S. and U.B. conceived the project and developed the experimental design. A.V., F.S. and J.L.P. carried out the fieldwork and data collection. A.V., M.C.W. and M.Z. carried out logistic regression analyses. A.V. and T.B. carried out neighbourhood asymmetry and allometric relations analyses, A.V., M.C.W., M.Z., E.D. and C.P discussed and performed network analyses. A.V. wrote the manuscript. M.C.W., T.B., C.P., J.L.P. and U.B. helped in revisions and manuscript edition.

## Competing interests

The authors declare no competing interests.
