## [Peer Review File · Communications Biology]

This manuscript has been previously reviewed at another Nature Research journal. This document only contains reviewer comments and rebuttal letters for versions considered at Communications Biology.

Reviewers' comments:

Reviewer #1 (Remarks to the Author):

I really appreciated the rebuttal letter from the authors to my comments, which demonstrated their deep understanding of the shortcomings of their study. However they failed to include their arguments and explanations into the paper itself! The first major point was that they did not confirm that the recorded root grafts were indeed "functional"; They offered a long discussion in their rebuttal letter and although they added that non functional root grafts still confer higher mechanical stability to trees, they should acknowledge the fact that functionality was not verified and arguments linked to nutrient transfers may not be true in this case. Same goes for their affirmation that root grafting benefitted tree growth or that salinity stress decreased root grafting; the methodology does not allow to demonstrate that tree growth increased following root grafting because the grafts were not aged and put into relation with tree growth. In the same way, because trees in high salinity areas are smaller, their roots have less chances to cross and form root grafts hence the lower frequency of root grafting could be a simple results of trees being smaller and not because of the higher stress gradient. I am not saying that what authors are suggesting is false, but that they have to acknowledge - in the paper - the possible shorthcomings of the study, as they did in their rebuttal letter.

P.3 "root-graft" or "root graft" but not both should be used throughout

P.3 "Non-functional grafts confer higher mechanical stability" is a little confusing because the reader could interpret it as if non-functional grafts confer more stability than functional grafts. Should be rephrased to say that "even if they are not physiologically functional, these graft can increase tree stability as compared to non grafted trees"

P.4 *Avicennia germinans* L. the authority (L.) should not be italicized

P.22 "root grafting slenderness"... shouldn't it be the effect of root grafting on slenderness ratio of trees?

Reviewer #2 (Remarks to the Author):

The authors study root grafts in the black mangrove, *Avicennia germinans*, which grows in stressful conditions composed of flooded and highly saline soils. They show that root grafts occur more in conditions of higher stress, and group size decreases in stressful conditions. As trees grow larger in diameter, they are more likely to form root grafts but the presence of stressful environmental conditions is correlated with trees grafting at smaller stem diameters. This is compelling evidence for the advantages of root grafts between individuals of the same species.

I think this study will spur much additional research in this field. There were concerns from previous reviewers that the authors could not verify that roots were truly grafted and functional, but I find that the authors have responded well to the previous reviewer's concerns and have clarified and strengthened the manuscript. This paper documents field observations of root grafts forming, and future research can build on this by examining physiological performance of trees with and without grafts; whether plant structure such as multiple stems affect grafts; and whether kinship affects grafting.

Reviewer #1 (Remarks to the Author):	Remarks: All changes made to the manuscript have been highlighted in blue to facilitate tracking by the reviewers.
Comment	Response
I really appreciated the rebuttal letter from the authors to my comments, which demonstrated their deep understanding of the shortcomings of their study. However they failed to include their arguments and explanations into the paper itself! The first major point was that they did not confirm that the recorded root grafts were indeed "functional"; They offered a long discussion in their rebuttal letter and although they added that non functional root grafts still confer higher mechanical stability to trees, they should acknowledge the fact that functionality was not verified and arguments linked to nutrient transfers may not be true in this case.	We are very grateful with the reviewer for all her/his comments to our work. We feel that it has significantly contributed to improve the quality and clarity of the manuscript and has helped us a lot in identifying shortcomings and, once again, the reviewer is right, we should have increased the clarity of our methods and limitations within the paper. We sincerely apologise for failing on that regard during the first round of revisions. We have now included much of the information discussed in the rebuttal letter in different sections of the manuscript. We break down the changes here: We added a paragraph [Lines 91-105] after discussing the advantages of working with mangrove ecosystems, where we also highlight the challenges of the research in regards to i) the uncertainty of growth rings within tropical forests, and ii) the compromises made between performing especially explicit studies of this nature in mangroves, where extensive extraction of grafts for whole stands for anatomical verifications was not viable. We further clarify and justify that- and why functionality of grafts was not verified [Lines 97-105] We have further added a “concluding remarks” section at the end of the results and discussion section, discussing more in depth the indirect evidence of positive interactions via root grafts based on our results. Here we highlight that cause-effects of root grafts on tree size cannot be inferred with our results and point out that the quantification of functional root grafts, as well as their size and age are detrimental to generate a mechanistic understanding of the processes regulating resource transfer withing networks, and their contribution to tree and forest stand performance [Lines 233-249]
Same goes for their affirmation that root grafting benefitted tree growth or that salinity stress decreased root grafting; the methodology does not allow to demonstrate that tree growth increased following root grafting because the grafts were not aged and put into relation with tree growth. In the same way, because	Both covariates (stand density and salinity) independently, showed a negative effect on P(grafting), this is correct, and it is now highlighted in the main document that this could be associated to trees being smaller and having less chances to cross with other tree roots [Lines 140-141]. Still, the interaction term between root grafting and density had a positive effect for small diameter trees (See Extended Data Table 1), while the highest proportion of grafting was reported for the site with highest density and salinity,

trees in high salinity areas are smaller, their roots have less chances to cross and form root grafts hence the lower frequency of root grafting could be a simple results of trees being smaller and not because of the higher stress gradient. I am not saying that what authors are suggesting is false, but that they have to acknowledge - in the paper - the possible shortcomings of the study, as they did in their rebuttal letter.	we have edited the last sentence of the paragraph for greater clarity [Lines 144-145]. We also re-worded the last sentence of the first paragraph of page 7 to ensure that the observed changes in tree allometry could either be due to potential increased growth rates (in case that grafts were functional) or just to increased mechanical stability. The further reiterate this in the “concluding remarks” section [Lines 240-244]
P.3 "root-graft" or "root graft" but not both should be used throughout	Resolved, we have made sure that the term “root graft” is used consistently all along the manuscript. Only one change was made [Line 57].
P.3 "Non-functional grafts confer higher mechanical stability" is a little confusing because the reader could interpret it as if non-functional grafts confer more stability than functional grafts. Should be rephrased to say that "even if they are not physiologically functional, these graft can increase tree stability as compared to non-grafted trees"	We appreciate the reviewer for bringing this possible misinterpretation to our attention, we have edited the sentence to “... Still, even if grafts are non-functional, they can increase tree stability as compared to non-grafted trees by sharing anchoring systems”. [Lines 63-64]. Please note that for or improved readability we did a minor editing to the whole paragraph [Lines 60-66]
P.4 Avicennia germinans L. the authority (L.) should not be italicized	We appreciate the reviewer for bringing this typing mistake to our attention, the authority font has now been corrected [Line 85]
P.22 "root grafting slenderness"... shouldn't it be the effect of root grafting on slenderness ratio of trees?	We sincerely apologise for this editing mistake, the reviewer is right and we have amended the sentence, it now reads “the effect of root grafting on tree slenderness ratio” [now found on Line 374].

Additionally, please note that the whole manuscript has been edited in line with Communications Biology Guidelines: References were removed from the abstract, manuscript headings “abstract” and “Results and Discussion” were added. References from the main text are now merged with the references of the methods section, and the reference list now appears after the “methods” section.

Reviewer #2 (Remarks to the Author):	
The authors study root grafts in the black mangrove, Avicennia germinans, which grows in stressful conditions composed of flooded and highly saline soils. They show that root grafts occur more in conditions of higher stress, and group size decreases in stressful conditions. As trees grow larger in diameter, they are more likely to form root grafts but the presence of stressful environmental conditions is correlated with trees grafting at smaller stem diameters. This is compelling evidence for the advantages of root grafts between individuals of the same species. I think this study will spur much additional research in this field. There were concerns from previous reviewers that the authors could not verify that roots were truly grafted and functional, but I find that the authors have responded well to the previous reviewer's concerns and have clarified and strengthened the manuscript. This paper documents field observations of root grafts forming, and future research can build on this by examining physiological performance of trees with and without grafts; whether plant structure such as multiple stems affect grafts; and whether kinship affects grafting.	Since no further recommendations were added, we can only send here our appreciation to all previous comments made by the reviewer in earlier versions of the manuscript. We hope that the publication of this study will stimulate further multidisciplinary research to better understand the cost/benefit and ecological (eco-physiological) implications of root graft formation in natural forests.

We sincerely hope that the reviewers agree with the improved presentation of the research question, results and discussion. We think that the aims of the study and its limitations are now well addressed and our methods are compensating the lack of anatomical verifications with comprehensive analyses of tree allometry, accounting for competition pressure and network structures to provide a wider perspective of root grafts on tree interactions, while carefully discussing the results based on our evidence and avoiding conclusions that would otherwise not be supported by our results.